# Impact of deep learning and post-processing algorithms performances on biodiversity metrics assessed on videos

Valentine Fleuré[1,2]*, Kévin Planolles[1,3], Thomas Claverie[4], Baptiste Mulot[2], Sébastien Villéger[1]

1 MARBEC, University Montpellier, CNRS, Ifremer, IRD, Montpellier, France, 2 ZooParc de Beauval & Beauval Nature, Saint-Aignan, France, 3 Research-team ICAR, LIRMM, University Montpellier, CNRS, Montpellier, France, 4 UMR ENTROPIE, IRD, IFREMER, CNRS, Univ La Réunion, Saint Denis, Réunion, France

* valentine.fleure@gmail.com

## Abstract

Assessing the escalating biodiversity crisis, driven by climate change, habitat destruction, and exploitation, necessitates efficient monitoring strategies to assess species presence and abundance across diverse habitats. Video-based surveys using remote cameras are a promising, non-invasive way to collect valuable data in various environments. Yet, the analysis of recorded videos remains challenging due to time and expertise constraints. Recent advances in deep learning models have enhanced image processing capabilities in both object detection and classification. However, the impacts on models' performances and usage on assessment of biodiversity metrics on videos is yet to be assessed. This study evaluates the impacts of video processing rates, detection and identification model performance, and post-processing algorithms on the accuracy of biodiversity metrics, using simulated remote videos of fish communities and 14,406 simulated automated processing pipelines. We found that a processing rate of one image per second minimizes errors while ensuring detection of all species. However, even near-perfect detection (both recall and precision of 0.99) and identification (accuracy of 0.99) models resulted in overestimation of total abundance, species richness and species diversity due to false positives. We reveal that post-processing model outputs using a confidence threshold approach (i.e., to discard most erroneous predictions while also discarding a smaller proportion of correct predictions) is the most efficient method to accurately estimate biodiversity from videos.

## Introduction

The escalating biodiversity crisis, exacerbated by climate change, habitat destruction, and exploitation, underscores the urgent need for efficient monitoring strategies of

**Data availability statement:** All data are fully available. Research data are available at https://doi.org/10.5281/zenodo.15519964. Code is available at https://github.com/valentine-fleure/deep_perf.

**Funding:** V. F. was supported by the Association Beauval Nature (CIFRE 2022/0127). K. P was supported by the IA-Biodiv ANR project FISH-PREDICT (ANR-21-AAFI-0001-01). The funders had no role in study design, data collection and analysis, decision to publish, or preparation of the manuscript.

**Competing interests:** The authors have declared that no competing interests exist.

biodiversity [1]. Such efficient monitoring requires consistently tracking the presence and abundance of species at large spatial scales, in various habitats and frequently to assess ecosystem changes accurately [2]. This challenge is particularly critical in underwater environments, where conducting biodiversity surveys is logistically complex and resource-intensive. Coral reefs, which alone host nearly one-fifth of all marine life on Earth [3], are among the most threatened ecosystems, making it essential to develop efficient monitoring techniques to assess and protect their biodiversity effectively. Various community assessment methods are conventionally used but none of them are yet satisfactory. Sampling-based approaches such as fishing and trawling observations yield valuable data, but they often come with ecological impacts [4]. Among non-invasive methods, eDNA allows censusing a high number of species (including elusive or cryptic ones) but it does not provide estimates of species abundance [5]. Acoustic based approaches allow long term monitoring and development of ecosystem health proxies, but do not census all species and do not accurately estimate abundance [6]. Visual counting methods performed by scuba divers are reference approaches to survey community composition but are notoriously time-consuming and require highly trained experts [7]. In response to these challenges, the use of video-based surveys has emerged as a promising solution [8] with diverse approaches depending on the animals and ecosystems studied with underwater camera traps [9], baited underwater stereo-cameras for marine predators [10,11], long-duration remote underwater cameras for reef fishes [12,13], autonomous underwater vehicles for reef fishes [14], and unmanned aerial vehicle for marine megafauna [15,16]. These image recorders allow for accurate and non-invasive monitoring of marine and terrestrial ecosystems and have thus been increasingly used for the last decade [8]. However, the on-screen analysis of recorded videos by experts remains a significant obstacle to scalability. The detection and identification of all targeted species are indeed demanding tasks even for highly-trained experts [17,18].

Meanwhile, the rise of the use of image recording and artificial intelligence applied to image processing have made significant progress during the last decade, mostly through the progress of deep learning algorithms [19]. Today, two of the main families of deep learning models applied to images are used: detection models and classification models. Detection models are used to find all objects of interest in an image [20]. Classification models are used to associate each image with a label [21]. While multi-class detection models do both tasks simultaneously [22,23], training them efficiently remains a challenge when most classes are rare, which is the case for most species assemblages [24–26]. Deep learning algorithms can identify all animals on images considerably faster than humans [27]. However, detection algorithms produce both false negatives (individuals that are not detected) and false positives (regions of image without any individuals but detected as containing one). In addition, identification algorithms may assign the wrong species to the correctly detected individuals. Finally, all false positives from the detection step, yield a misidentification as classifiers always attribute one of the learned classes to an input object. One approach to limit model errors consists in thresholding [28], i.e., setting aside raw outputs

according to a user-defined criterion. This criterion can be based on a confidence score returned by the models [29] or on the size of the bounding box [30].

Computer vision models are designed to process images [31]. Hence, to process videos the first step is to extract a subset of the video frames after choosing a frame rate (e.g., 1 image per second among the 25 or 30 images per second recorded). Then, models will analyze these images one by one. The selected frame processing rate affects the number of images analyzed hence the processing time as well as the potential to accurately assess biodiversity. Increasing the processing frame rate is expected to reduce false negatives from the detection model (more frames will increase the chance to catch an individual) and hence allow censusing more species and more individuals. Meanwhile, an increasing processing rate is also expected to increase the number of false positives from both detection and classification models resulting in biased species richness and species abundances. However, the impact of frame processing rate on quality of detection and identification outputs is yet to be assessed.

Furthermore, the ultimate goal of such automated video processing is to compute biodiversity metrics, such as total abundance, species richness, and species diversity, based on outputs from the models on the sequence of images from the video. In more detail, the species richness will be accurately estimated as long as at least one individual of each species is correctly detected and classified in at least one frame of the video, with the condition that none of the detections are classified as a species that does not appear in the video. Total abundance estimated using the maximum number of individuals present simultaneously in a single image (i.e. maxN) can be accurately estimated only if for each species all individuals are detected and identified on at least one of the images where the maximum number is visible [32]. This metric is highly sensitive to detection, as only a few frames typically contain the maximum number of individuals, hence false negatives prevent from censusing all individuals. It is also sensitive to classification, since misclassifications (i.e., false negative) in those key frames decrease the number of individuals from a given species seen simultaneously. Hence, some biodiversity metrics are by definition more sensitive to model errors, yet such differences remain unevaluated to date.

In the present study, we evaluate the influence of video processing rates, the performance of detection and identification models, and the use of post-processing thresholds on the accuracy of biodiversity metrics estimation using simulations of remotely censused communities. We achieve this using simulated remote videos of fish communities and simulated 14,406 automated processing pipelines, and calculating error rates and biodiversity metrics.

## Materials and methods

### Simulating video-based community surveys

We considered a hypothetical case study with a regional pool of 30 species (close to the richness observed for vertebrates in terrestrial and aquatic temperate ecoregions) [24,33,34].

The 30 species were divided into 8 groups with contrasting abundance and temporal abundance variation in a site (Table 1). More precisely, groups were characterized by varying: (i) abundance, defined as the maximum number of individuals seen simultaneously (ranging from 1 to 20); (ii) relative temporal occupancy, or the proportion of time they were present (ranging from 0.5% to 90%); and (iii) the clustering of their presence across time, represented by the number of time slots (from 1 to 15) when a species was visible.

These groups were designed to represent the variability of abundance, mobility (i.e., average moving speed and proportion of time spent stationary for feeding or resting), and gregariousness observed in most animal communities surveyed with remote cameras (e.g., solitary fox vs gregarious mouflon, gregarious surgeonfishes vs solitary barracuda). We note that the scarcest and most elusive species is represented by a single individual recorded for only half a second over 10 minutes while the most abundant species is represented by 20 individuals visible for 2 minutes over 10 minutes and the most present species is represented by 2 individuals visible for 9 minutes over 10 minutes (S1 Fig). The number of species in each group varied from 3 to 5 species (Table 1).

**Table 1. Values used to describe the groups of species considered in this study.**

| group | abundance | relative temporal occupancy (%) | number of time slots | number of species in the regional pool | number of species in each 10-min video |
|---|---|---|---|---|---|
| 1 | 20 | 20 | 5 | 3 | 1 |
| 2 | 2 | 90 | 15 | 3 | 1 |
| 3 | 2 | 30 | 15 | 3 | 2 |
| 4 | 8 | 10 | 1 | 3 | 2 |
| 5 | 8 | 5 | 1 | 3 | 2 |
| 6 | 1 | 5 | 1 | 5 | 2 |
| 7 | 1 | 1 | 1 | 5 | 3 |
| 8 | 1 | 0.5 | 1 | 5 | 2 |

We simulated fixed remote video surveys with a total duration of 10 minutes. This length was chosen as a trade-off between the ability to simulate contrasted temporal abundance of species and the size of simulated datasets.

We considered that each video survey allows recording 15 species (out of the 30 from the regional pool) representing the 8 groups with 1–3 species, sorted randomly (Table 1).

Then, the simulation of abundance distribution through time of each species was led through 4 steps. First, the average duration of a presence slot was computed as the ratio between temporal occupancy (min) and the number of presence slots. Second, the duration of each presence slot was sorted from a normal distribution with the mean being the average presence time and the standard error as the square root of the mean time. Third, the time interval between two consecutive presence slots was iteratively sorted from a uniform distribution, ranging from the number of remaining time slots to the total remaining time. Lastly, the number of individuals within each presence slot was sorted from a binomial distribution with the duration of the time slot, incorporating the species' abundance.

These temporal abundances were simulated with a resolution of one second, which corresponds to the unit of measurement usually used during diver counts. They were reduced to 30 frames per second to simulate video recording at the site studied.

To demonstrate the robustness of our results, 10 video surveys were simulated to account for variability in species abundance through time. The number of replicates was limited to 10 to balance computational cost with reproducibility.

## Simulation of video processing with detection and identification algorithms

For each video, we simulated an automated processing with a detection algorithm and then a classification algorithm with varying performances. For the detection model, we used recall to assess false negatives, ensuring that all present individuals are detected, and precision to evaluate false positives, minimizing incorrect detections. For the classification model, we relied on accuracy to measure the proportion of correctly classified images, providing an overall assessment of classification performance. This approach follows standard evaluation practices, where recall and precision are essential for detection tasks, while accuracy is commonly used for classification.

We first varied the processing rate of video processing with 7 levels from 0.25, 0.5, 1, 2, 5, 10, and 30 frames per second. The two first processing rates mean 1 image processed every 4 or 2 seconds, respectively. The last level is the processing of all the frames from a standard video recording. The subsampling consisted in keeping the first frame of each second, then evenly choosing among the next ones depending on the target processing rate.

The first step of the automated analysis was the detection of all individuals using a single-class detection algorithm. The recall measures the ability of a model to identify all relevant positive instances. The recall score of the algorithm varied from 0.60 (similar to the least efficient algorithm published) to 0.99 (i.e., hypothetically quasi-perfect algorithm) with 5 intermediate levels (0.70, 0.80, 0.85, 0.90, 0.95). To simulate this, each individual present in the image has a detection

probability calculated using a Bernoulli distribution with recall as a probability. False negatives, i.e., undetected individuals, are not treated in the rest of the simulations as will be the case in real-life pipelines.

The precision is the proportion of correctly labeled individuals among all those detected as present. The precision score of the algorithm also varied from 0.60 to 0.99 with 5 intermediate levels (0.70, 0.80, 0.85, 0.90, 0.95). We determined the number of false positives using the precision rate and the number of detections, then randomly added these false positives across all frames. We acknowledge that in real-world applications, false positives could be clustered, typically caused by recurring background features that resemble target objects.

Second, a species-level classification algorithm was simulated for each bounding box from the detection algorithm. The accuracy is the proportion of correctly labeled individuals among all individuals of this class. The algorithm had an accuracy score ranging from 0.60 to 0.99 with 5 intermediate levels (0.80, 0.70, 0.85, 0.90, 0.95). We thus simulated misidentification at random using 1-accuracy as a probability of misidentification and the identity of the erroneous species was randomly chosen among the remaining species from the regional pool (i.e., 29 species). We hence assume that the misidentifications were even across species (i.e., there was no species pair more likely to be confounded to each other). We acknowledge that in real-world applications misidentifications are often not random with on average lower accuracy for the classes with the fewer images in the train set and across classes with similar visible features.

Third, we implemented a post-processing step inspired by the confidence threshold approach of Villon (2020) that transfers some identifications (expected to be mostly misidentifications) to an unsure class. For each misidentification, there was a probability $p$ (5 levels: 0.80, 0.85, 0.90, 0.95, 0.99) that it was eventually considered as "unsure". Meanwhile, for each good identification, there was a probability $p/20$ that it was eventually considered as "unsure". We also analyze results from detection and identification steps without this post-processing.

We thus simulated a total of 14,406 automated processing pipelines (7 processing rates * 7 detection recall * 7 detection precision * 7 classification accuracy * 6 post-processing thresholds) for each of the 10 simulated videos.

For each processing pipeline, we eventually computed the abundance of each species as the maximum number of individuals present in a single frame (i.e., MaxN metric commonly used in video-based surveys).

## Assessing the performance of automated processing to estimate biodiversity metrics

As a preliminary analysis, we checked whether all individuals were detectable (i.e., present on at least one of the images) on the frames kept for analysis with each processing rate.

We assessed the influence of algorithms' performance on three biodiversity metrics frequently used in the monitoring of ecosystems.

First, we computed total abundance as the sum of species abundances. Second, we computed species richness as the number of species seen along the video (i.e., at least one individual in at least one frame). Third, we computed species diversity using the Hill number version of the Shannon entropy [35]: $exp\left(-\sum_{i=1}^{S} p_i \times log\left(p_i\right)\right)$, where $S$ is the total number of species present and $p_i$ is the abundance of species $i$ relative to the total number of individuals. Species diversity increases with increasing number of species and increasing evenness of their abundances.

These 3 indices were computed for data from ground truth (simulated communities) and for each output of each processing pipeline (species abundances provided by detection, classification, and post-processing algorithms).

To further investigate the outputs of the processing pipeline we computed Jaccard and Bray-Curtis similarity indices between species composition or abundance from each processing pipeline and species composition or abundances from ground truth.

The Jaccard index [36] quantifies the similarity between two sets of species by dividing the number of shared species by the total number of distinct species across both sets. It ranges from 0 (no species shared) to 1 (all species shared).

The Bray-Curtis similarity [37], quantifies differences between two sets of species abundances. it is defined as follows: $\frac{2 \times \sum_{i=1}^{p} min\left(N_{ij}, N_{ik}\right)}{\sum_{i=1}^{n} \left(N_{ij} + N_{ik}\right)}$, where $N_{ij}$ is the number of individuals of species $i$ at site $j$, $N_{ik}$ is the number of individuals of species $i$ at site $k$, and $p$ is the total number of species in the samples. It ranges from 0 (sets have different species composition or abundance markedly differ for shared species) to 1 (all species have the same abundance in both sets).

## Results

All 15 species simulated on each video were detectable (i.e., present on at least 1 of the images) for analyses with a processing rate of 1 image per second or higher. For the 2 lowest processing rates (0.25 and 0.5 images per second), 14.3 (sd = 0.67) and 14.8 (sd = 0.42) species on average were detectable, respectively (S1 Table).

For total abundance, the ground truth was 65 individuals present in each simulated community. Estimated abundance increased with the frame processing rate for all combinations of deep learning model performance, reaching a maximum of 187 individuals for a detection model with a recall of 0.99 and a precision of 0.6 (fifth row on Fig 1), and an identification model with an accuracy of 0.6 (red dot on Fig 1).

For all processing pipelines for which the processing frame rate was 2 and above, the estimated number of species was 30, which corresponds to the number of species learned by the classification model. For lower processing rates (0.25, 0.5 or 1), estimated species richness was lower than 30 and even reached 16 for a recall of 0.6, a precision of 0.99 and an accuracy of 0.99.

Species diversity was 8.71 for the ground truth. For all combinations of model parameters, the estimates of this metric were all above the ground truth and increased with the processing frame rate. It was at a minimum of 10.0 for a detector with a recall of 0.6 and a precision of 0.99 (second row on Fig 1) and a classification model with an accuracy of 0.99 (pink dot on Fig 1) for a processing frame rate of 0.25. Estimated diversity even reached a maximum of 27.8 for a detector with a recall of 0.6 and a precision of 0.6 (top row on Fig 1) and a classification model with an accuracy of 0.6 (red dot on Fig 1) for a processing frame rate of 30.

The Jaccard similarity index between actual species composition and those estimated after automated processing was constant at 0.5 except for 36 models with a processing rate of 0.25 or 0.5 frames per second, for which the Jaccard similarity was between 0.32 and 0.49 (S2 Fig).

The Bray Curtis similarity between actual abundances and those estimated after automated processing decreased with increasing processing rate from 0.72 for 1 frame processed by second to 0.64 for 30 frames processed by second (Fig 2).

Given these results, we focus hereafter on the effects of algorithm performance for a processing rate of 1 image per second to avoid missing out on rare species.

For total abundance, using a 0.99 confidence threshold post-processing, it was possible to get closer to the ground truth for all the detection models. A detection model with a recall of 0.6 and a precision of 0.6 (top row on Fig 3) estimated 7% more abundance than the ground truth, regardless of the accuracy of the classification model (colors of dots on Fig 3), when the post-processing threshold was between 0.8 and 0.95.

For a detection model with a precision of 0.6 (first and third rows on Fig 3) the post-processing threshold had no impact on the richness for a threshold under 0.95. Setting threshold above 0.99 led the species richness being closer to the ground truth [15], averaging 23.9 for a threshold of 0.99. For detection models with a precision of 0.99, the impact of threshold post-processing was greater when the accuracy of the classification model was close to 1 (Fig 3).

The Jaccard similarity between actual species composition and those estimated after automated post-processing decreased regardless of the classification model performance (S3 Fig). With a post-processing threshold of 0.99, the maximum increase was 0.45, going from 0.51 to 0.96 for a model with a recall of 0.99, a precision of 0.99, and a classification accuracy of 0.99.

Using a post-processing threshold increased the Bray-Curtis similarity (Fig 4) between actual abundances and those estimated after automated processing for all performances of detection or classification models. With a post-processing

   

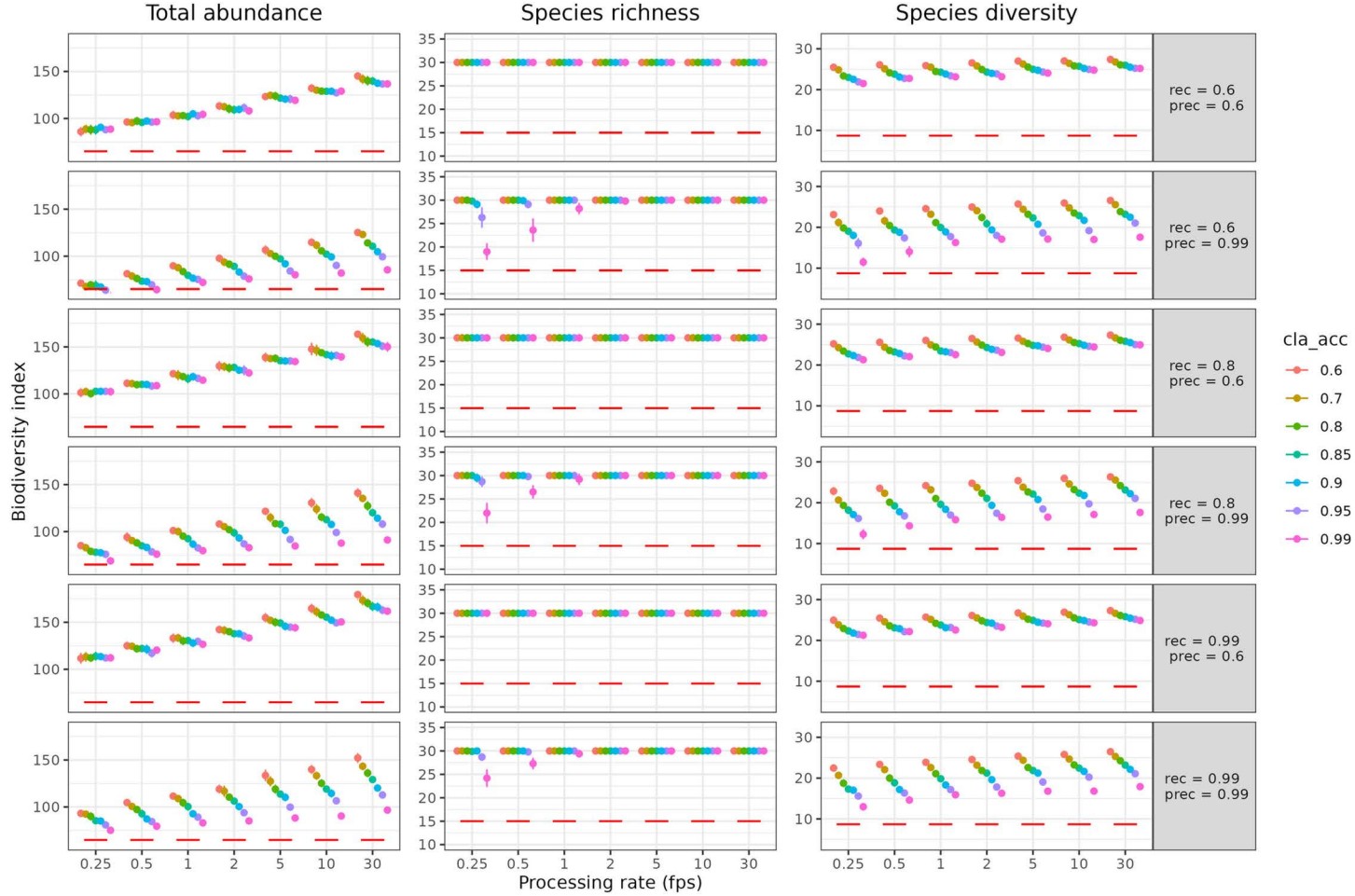

**Fig 1. Effects of processing rate and algorithms performance on estimates of biodiversity.** *Total abundance, species richness and species diversity estimates as a function of video processing rate (fps, frame per second) for 42 automated analysis models resulting from 6 detection models, with their respective recall ("rec") and precision ("prec") performances (rows) and 7 classification models (accuracy ("cla_acc") as colors). Each dot represents the average over the 10 simulated videos with corresponding standard error as vertical bars. The red horizontal bar represents the ground truth (i.e., diversity facet present on each video). Total abundance is the sum of the maxN (maximum number of individuals of a species seen in one image) of each species, richness is the number of species seen, and species diversity is the exponential of the Shannon entropy index.*

threshold of 0.8, the minimum augmentation was 0.003, going from 0.791 to 0.794 for a model with a recall of 0.6, a precision of 0.95 and a classification accuracy of 0.99. With a post-processing threshold of 0.99, the maximum augmentation was 0.24, going from 0.61 to 0.85 for a model with a recall of 0.99, a precision of 0.6 and a classification accuracy of 0.6.

Although only 10 simulations were performed, the results were highly consistent across replicates, as indicated by the very low standard deviations around the mean values shown in the figures.

## Discussion

In our simulations, the scarcest species were not always detectable by the models processing an image every 2 seconds (processing frame rate = 0.5) or an image every 4 seconds (processing frame rate = 0.25). Such species moving briefly across the camera field-of-view, such as fast-swimming predators (tuna, trevally) in the marine environment contribute

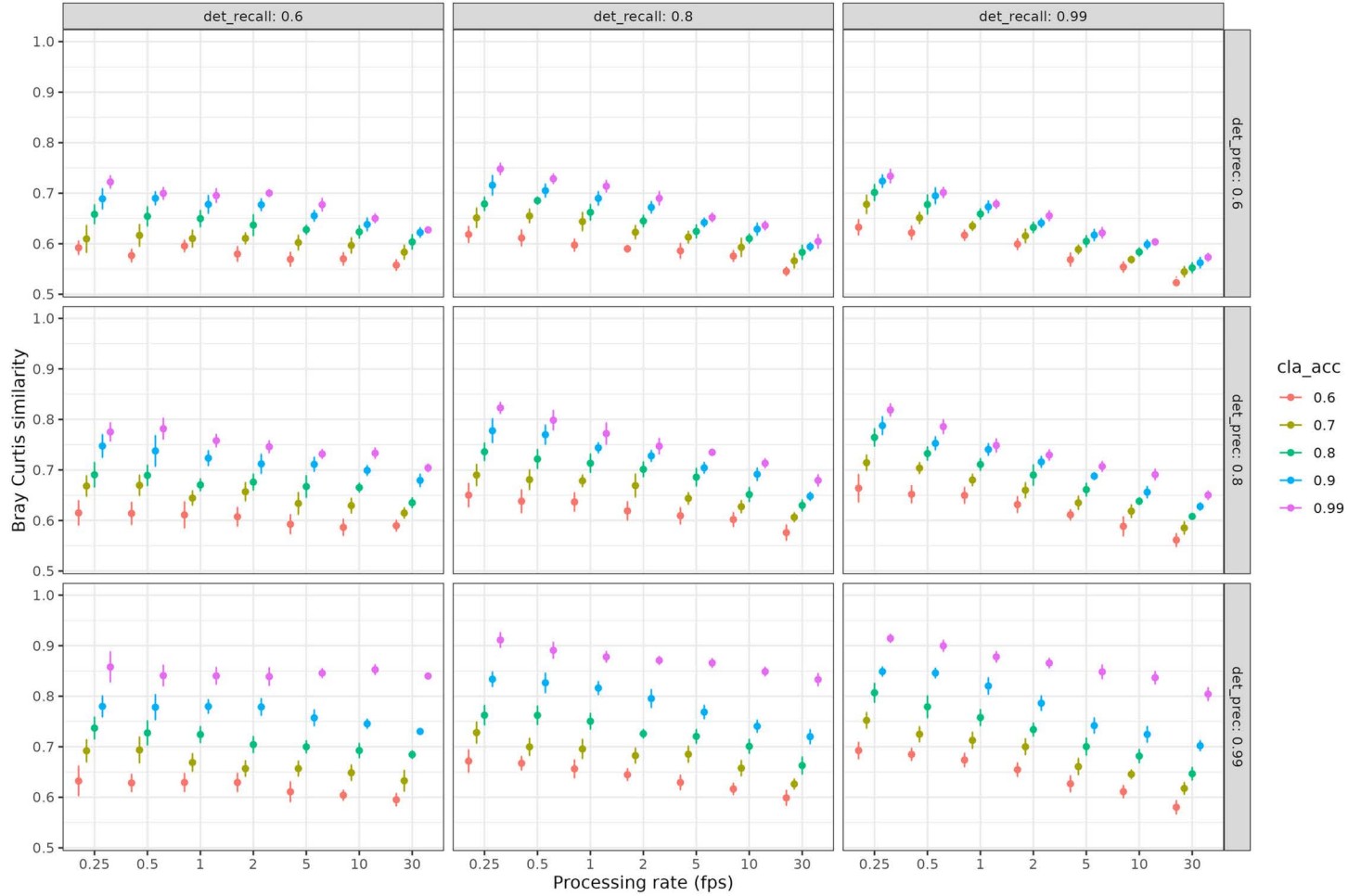

**Fig 2. Effect of video processing rate and algorithms performance on the estimation of the abundance-structure of simulated communities.** *Estimation of abundance-structure as a function of video processing rate (fps, frame per second) for 45 automated analysis models resulting from 9 detection models, with their respective recall ("det_recall" – columns) and precision ("det_prec" – rows) performances and 5 classification models (accuracy ("cla_acc") as colors). Each dot represents the average over the 10 simulations with corresponding standard error as vertical bars. The Bray-curtis similarity measures the difference in species abundances estimated after automated processing using algorithms and species abundances actually visible on videos.*

more than 10% to the species richness [18] and play important functions for ecosystems [38]. Hence, in most studies, processing rates should at least be 1 image per second to detect scarce species.

Eventually, with all processing rates above 1 frame per second, all species were detected by detection models. Regardless of the performance of the detection models, for processing rates of 0.25 and 0.5 image per second, only the rare species present < 1% of the time, were not detected. Hence, overall, the false negatives from the detection model (i.e., individual missed on images) did not markedly affect the estimated composition of the assemblage. It is important to note that although errors in detecting rare species at the level of individual frames may result in underestimating their presence, combining observations from multiple frames typically diminishes this bias when assessing broader ecological indicators. Consequently, despite the occurrence of detection errors at the frame level, their overall influence on biodiversity metrics can be reduced through repeated sampling. This partly accounts for our simulations indicating a general overestimation rather than underestimation of species richness.

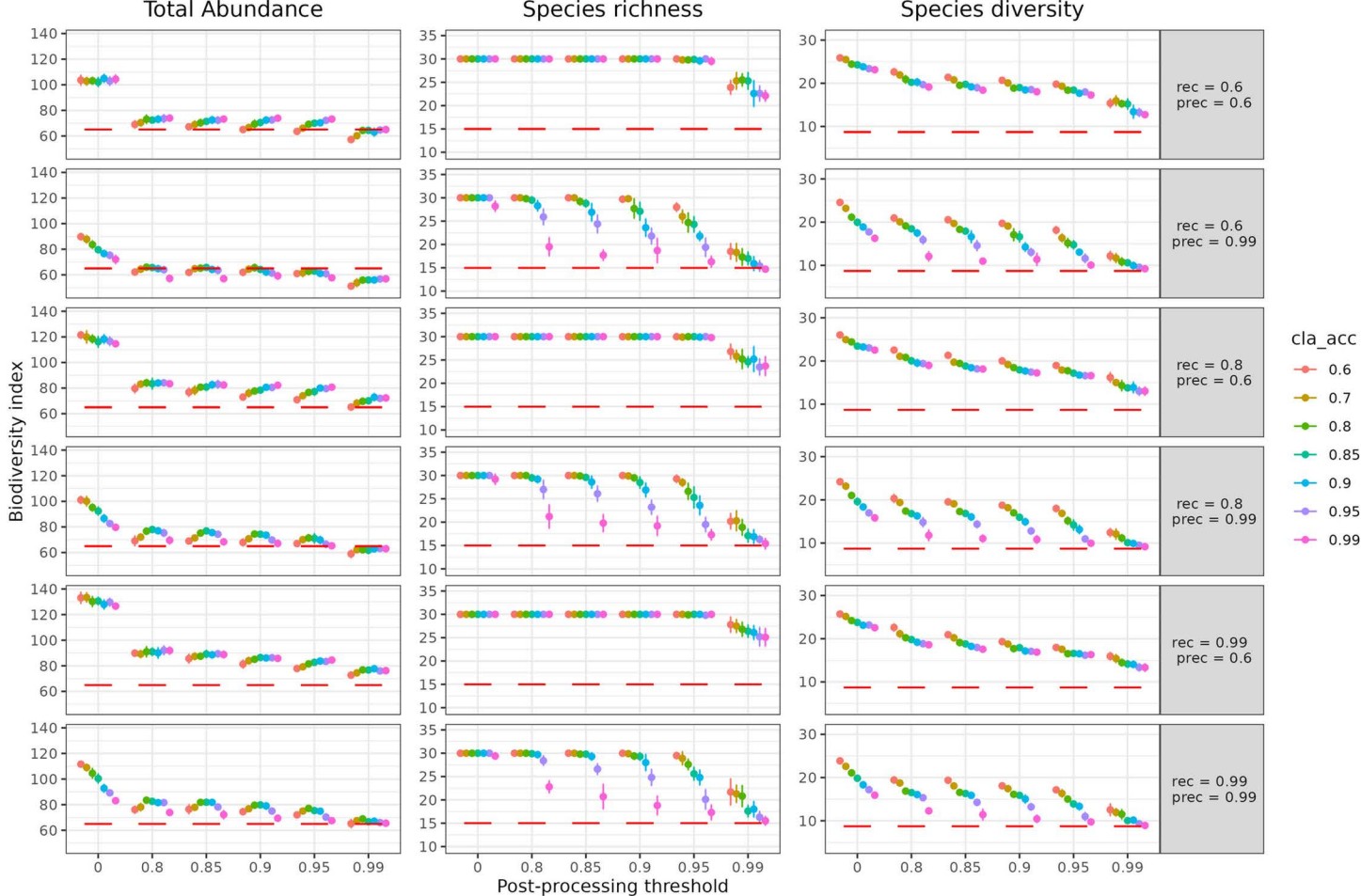

**Fig 3. Effect of post-processing model outputs on biodiversity estimates.** *Total abundance, species richness and species diversity estimates as a function of post-processing threshold for 42 automated analysis models resulting from 6 detection models, with their respective recall ("rec") and precision ("prec") performances (rows) and 7 classification models (accuracy ("cla_acc") as colors). Each dot represents the average over the 10 simulated videos with corresponding standard error as vertical bars. The red horizontal bar represents the ground truth. Total abundance is the sum of the maxN (maximum number of individuals of a species seen in one image) of each species, richness is the number of species seen, and species diversity is the exponential of the Shannon entropy index. Post-processing applies a confidence threshold to outputs of the identification models, discarding those with the lowest confidence scores to minimize misidentifications.*

The near-perfect detection model (both recall and precision of 0.99) yielded on average 53 false positives (i.e., objects being actually part of the background) for 5,295 detectable fish when the 10-minutes videos were processed at 1 frame per second. There were up to 527 false positives for 52,927 detectable fish when videos were processed at 10 frames per second and 1,586 false positives for 158,789 detectable fish when processed at 30 frames per second. According to our simulation design, the number of false positives is proportional to the number of frames analyzed, which is realistic as false positives occur mostly because of textured sessile organisms (e.g., algae, corals, sponges) [39]. Therefore, we recommend using a processing frame rate of 1 frame per second for video analysis, in order to ensure all species are detected while minimizing the number of false positives. However, if fast-moving species are not present or are not in the scope of monitoring, it is possible to analyze fewer images per second. In such contexts, using a lower processing rate (e.g., 0.5 fps) does not compromise species detection, while significantly reducing the number of analyzed frames and the number of false positives, thereby reducing the proportion of error in biodiversity estimates.

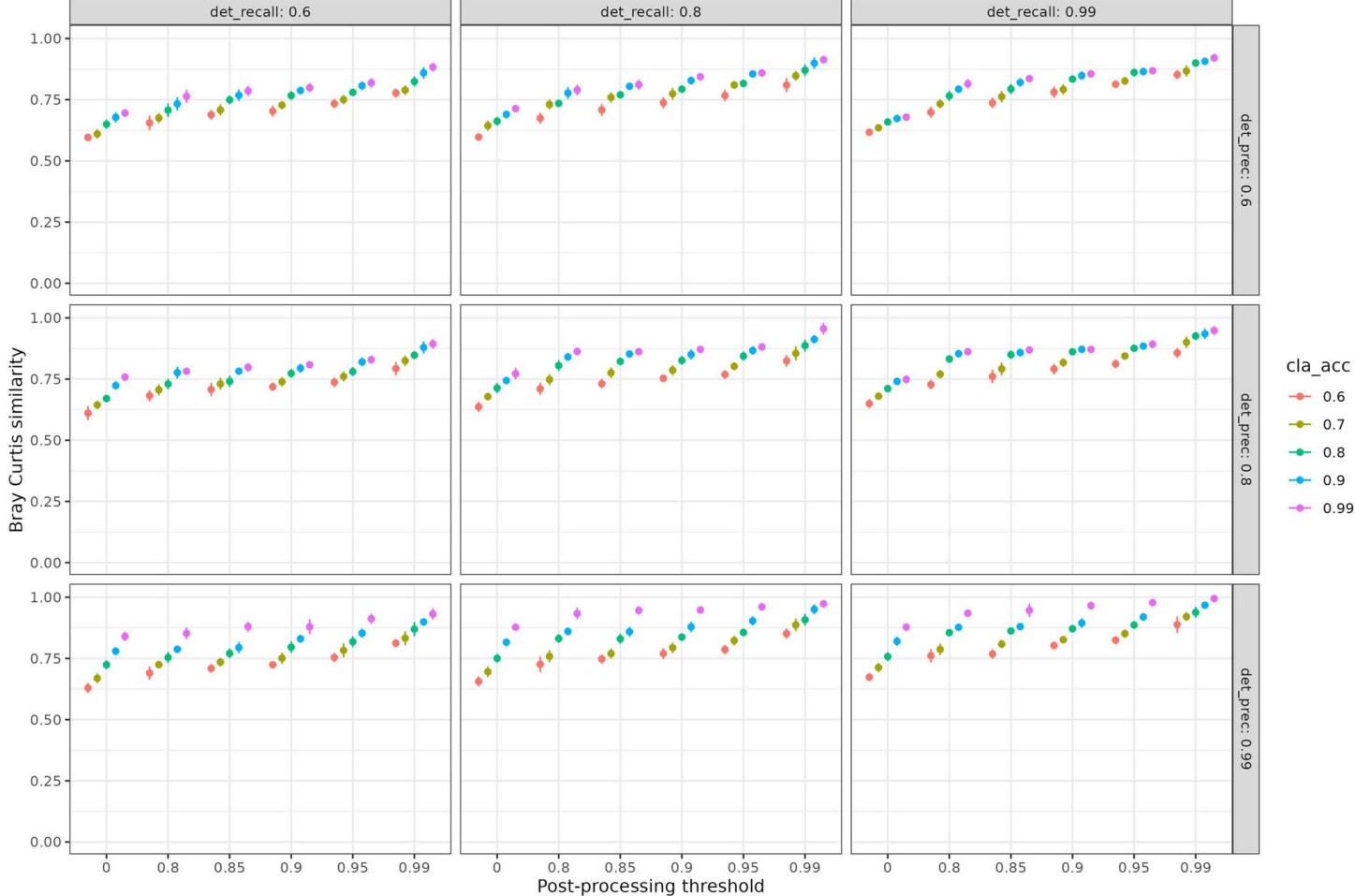

**Fig 4. Effect of post-processing threshold and algorithms performance on the communities abundance-structure estimation.** *Estimation of abundance-structure as a function of post-processing threshold for 45 automated analysis models resulting from 9 detection models, with their respective recall ("det_recall" – columns) and precision ("det_prec" – rows) performances and 5 classification models (accuracy ("cla_acc") as colors) for 1 frame per second processing rate. Each dot represents the average over the 10 simulations and the error bars are shown. The Bray-curtis similarity measures the difference in species abundances estimated after automated processing using algorithms and species abundances actually visible on videos. Each panel gathers results for a detection algorithm (recall in column and precision in row) and a classification algorithm (color corresponding to its accuracy). Post-processing applies a confidence threshold to outputs of the identification models, discarding those with the lowest confidence scores to minimize misidentifications.*

 

All 30 species present in the environment were always detected and identified when analyzing 1 frame per second, regardless of the recall and precision scores of the detection model, and the accuracy of the classification model. There were hence 15 false positive species, arising from classification errors, including those from false positives from the detection model. However, even with a near-perfect detection model (recall and precision = 0.99) and a near-perfect classification model (accuracy = 0.99), there were on average 29.4 (sd = 0.7) species identified at the end of the processing pipeline. With a near-perfect classification model (accuracy = 0.99), the errors were mostly due to false positives in the detection models. Indeed, background objects erroneously detected as objects of interest only lead to a misclassification, likely evenly picked among the list of putative species (i.e., present in the training set of the classification algorithm). In our simulations, we considered that errors in the confusion matrix were distributed randomly. In real cases, there are some

species pairs that are confused more than others (e.g., species with similar shapes and colors, or species with the most images in the training set). In such cases the final error in species richness assessment would depend on the number and co-occurrence of these confounded species: if they are often co-occurring (i.e., present in the same habitats), confusion will not lead to bias in richness spatial patterns, but if they have different habitats, the species richness will be inflated in all habitats (i.e., the 2 species will be erroneously detected everywhere). Therefore, as there were a large number of false positives all species were eventually erroneously estimated to be present. If classification of false positives from the detection tend to be towards a small subset of species (e.g., to cryptic species because of their similarity with benthos), such overestimation of species occurrence could be lower, but will still be an issue for biodiversity monitoring. One way to limit errors for these species is to apply a filter based on the size of the detected objects. While our study may not fully represent all the real-world complexities, such as non-random error patterns or unique community structures, the restriction in computational resources limited the scope of scenarios we could explore. However, the simulation code is openly available and can be adapted to test additional or more complex cases.

In addition to biases in species occurrence, and hence richness of communities, detection and classification models can lead to biases in species abundance. Here we found that total abundance was overestimated by 160% when video was processed at 1 image per second by detection and classification models with low performance (recall and precision of detection both <0.8 and classification accuracy <0.8). Even with a near perfect detection model (recall and precision both 0.99) and a near perfect classification model (classification accuracy = 0.99), abundance was still overestimated by at least 128% due to false positives.

Similarly, species diversity, which accounts for the evenness of species abundances, was overestimated by all models although less than species richness.

This lower sensitivity of abundance-based metrics to model errors resulted from two potentially cumulative patterns. First, a species' abundance is overestimated only if a false positive of this species is added to one of the video frames where this species was the most abundant. Second, false negatives due to detection or classification models could compensate for false positives.

Importantly, our simulations revealed that post-processing outputs from the identification model markedly improved estimation of the three biodiversity metrics (Fig 3, S3 Fig).

A detection model with moderate performance (precision and recall at 0.6) and a moderate classification model (accuracy at 0.6) with a post-processing threshold at 0.99, which overestimated species richness by 160%, outperformed a near-perfect detection model (precision and recall at 0.99) and a near-perfect classification model without post-processing, which overestimated species richness by 200%. Furthermore, the pipeline with models having moderate performance and a strict post-processing provided more accurate estimates of total abundance and species diversity, with error of 38% and 82% respectively, than the pipeline with high-performance models without post-processing (error of 95% and 74%) (Fig 3). This comparison highlights the importance of implementing post-processing of model outputs based on a confidence threshold.

For all detection models with a given precision, the gain with post-processing was the same (Fig 4). Indeed, for a model with a recall of 0.8, Bray Curtis similarity increased from 0.83 to 0.93 with post-processing, whereas for a recall of 0.9, Bray Curstis similarity increased from 0.83 to 0.97 with post-processing. We therefore recommend improving precision rather than recall for a detection model when using a post-processing on the outputs of the classification model. This aligns with previous works (e.g., [40]), which showed that conventional metrics like classification error may not fully capture a model's ability to reliably estimate key ecological indicators.

The post-processing method used in this study is similar to the one proposed by Villon et al. [26] which requires only splitting the image dataset into 2 subsets, one for training the model and one for setting the confidence threshold. Given our findings, we recommend setting the threshold to 0.99 to minimize the number of false positives and misclassifications. This aligns with previous works (e.g., [41]), which showed that applying threshold on classification outputs improve

accuracy, particularly for rare classes. However, this comes at the cost of reduced recall, which may lead to nondetection of some rare species, a trade-off that should be carefully considered depending on the monitoring goal.

Findings from our simulations involving a succession of two models — detection and classification — are also valid with a detection model alone or a classification model alone. A detection model alone serves as a processing pipeline where an expert performs classification with near-perfect accuracy (0.99), while a classification model alone represents detection by a human expert with near-perfect precision and recall (both 0.99). Multiclass detection models [23] provide an alternative to a single-class detection followed by a classification approach [22], sharing an overall recall for the detection and the classification and species-specific accuracy, thus aligning with our approach and conclusions. However, post-processing these models is challenging due to difficulties in independently training them twice to establish confidence thresholds for all species, especially rare ones with limited images [29]. Moreover, classification accuracy in multiclass detectors is generally lower compared to separate-step pipelines, which reinforces current recommendations to separate detection and classification tasks [26].

In conclusion, selecting a processing rate of 1 image per second optimizes the trade-off of minimizing false positives and subsequent misidentifications while ensuring the detection of rare species. Using a detection model with a high recall then applying a post-processing on outputs of the model classification using a strict confidence threshold eventually allows faithful estimates of key biodiversity metrics. Overall, these results demonstrate that automated processing of video with deep learning models having high, although not perfect performances, will ease the assessment of disturbance gradients and the benefits of protection on biodiversity.

## Supporting information

**S1 Fig. Example of the distribution of species abundances within a video.** The number of individuals of each species is illustrated with shades of green (white slots indicate absence of species).
(TIF)

**S1 Table. Number of detectable species according to the processing rate of videos.** Values are the number of simulations (out of 10) yielding 12, 13, 14, or 15 detectable species (i.e., at least 1 individual present on at least one frame). For each of the 7 processing rates (frames per second).
(DOCX)

**S2 Fig. Effect of video processing rate and algorithms performance on the estimation of species composition of simulated communities.** Estimation of the species composition as a function of video processing rate (fps, frame per second) for 45 automated analysis models resulting from 9 detection models, with their respective recall ("det_recall" – columns) and precision ("det_prec" – rows) performances and 5 classification models (accuracy ("cla_acc") as colors). Each dot represents the average over the 10 simulations with corresponding standard error as vertical bars. The Jaccard similarity index was used to measure the difference in composition of species present on the videos and the composition estimated after automated processing using algorithms.
(TIF)

**S3 Fig. Effect of post-processing threshold and algorithms performance on the species composition estimation.** Estimation of the species composition as a function of post-processing threshold for 45 automated analysis models resulting from 9 detection models, with their respective recall ("det_recall" – columns) and precision ("det_prec" – rows) performances and 5 classification models (accuracy ("cla_acc") as colors) for 1 frame per second processing rate. Each dot represents the average over the 10 simulations and the error bars are shown. Each panel gathers results for a detection algorithm (recall in column and precision in row) and a classification algorithm (color corresponding to its accuracy). Post-processing applies a confidence threshold to outputs of the identification models, discarding those with the lowest

confidence scores to minimize misidentifications. The Jaccard similarity index was used to measure the difference in composition of species present on the videos and the composition estimated after automated processing using algorithms. (TIF)

## Acknowledgments

The authors thank the two anonymous reviewers who helped improve and clarify this manuscript.

## Author contributions

**Conceptualization:** Valentine Fleuré, Sébastien Villéger.

**Methodology:** Valentine Fleuré.

**Visualization:** Valentine Fleuré.

**Writing – original draft:** Valentine Fleuré.

**Writing – review & editing:** Valentine Fleuré, Kévin Planolles, Thomas Claverie, Baptiste Mulot, Sébastien Villéger.

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
