## [Decision Letter · Decision Letter 0]

14 Apr 2025

Dear Dr. Fleuré,

Thank you for submitting your manuscript to PLOS ONE. After careful consideration, we feel that it has merit but does not fully meet PLOS ONE’s publication criteria as it currently stands. Therefore, we invite you to submit a revised version of the manuscript that addresses the points raised during the review process.

We look forward to receiving your revised manuscript.

Kind regards,

Tzen-Yuh Chiang

Academic Editor

PLOS ONE

Journal Requirements:

“V. F. was supported by the Association  Beauval Nature (CIFRE 2022/0127).

K. P was supported by the IA-Biodiv ANR project FISH-PREDICT (ANR-21-AAFI-0001-01).”

4. We notice that your supplementary figures are uploaded with the file type 'Figure'. Please amend the file type to 'Supporting Information'. Please ensure that each Supporting Information file has a legend listed in the manuscript after the references list.

Reviewers' comments:

Reviewer's Responses to Questions

**Comments to the Author**

1. Is the manuscript technically sound, and do the data support the conclusions?

Reviewer #1: Yes

Reviewer #2: Partly

2. Has the statistical analysis been performed appropriately and rigorously?

Reviewer #1: Yes

Reviewer #2: Yes

3. Have the authors made all data underlying the findings in their manuscript fully available?

Reviewer #1: Yes

Reviewer #2: No

4. Is the manuscript presented in an intelligible fashion and written in standard English?

Reviewer #1: Yes

Reviewer #2: Yes

Reviewer #1: General comments – I enjoyed reading this manuscript. A straightforward study processing simulated videos in different ways to calculate biodiversity metrics and compare them, to make some clear recommendations on post-processing for accurate measurements (e.g. 1 frame per second). In general, the study will be strengthened by situating the findings more in the relevant literature considering AI error rates and ecological metrics. I also think N = 10 videos per condition is a small number, although note the patterns between conditions do seem clear even with this sample size.

Introduction - Nice intro motivating the need for underwater video surveys, and the necessity/application of AI processing. There is a reasonable link made between AI errors and downstream effects on biodiversity indices, but, the intro would significantly benefit from a clearer explanation of how errors arise and are handled. A few unclear sentences to be reworded.

Line 50 – 53. Long sentence and a bit unclear. Add comma “without impact, but”. Can you clarify what is meant by structure vs. composition?

Line 57. Community not communities

Line 82 – Refer to this post-processing as thresholding to be consistent with the existing literature and expand on why this limits errors or cite a source which does so e.g. Improving the integration of artificial intelligence into existing ecological inference workflows - Cowans - Methods in Ecology and Evolution - Wiley Online Library which links AI error rates to ecological inference

Line 89 – “The selected frame processing rate affects the number of images analyzed and hence the likelihood to detect all species for a given model performance.” – perhaps also add considerations of time/processing efficiency here.

Lines 90 - 93 – This would be improved if you could clarify the types (false negative, false positive) and sources (animal missed, animal mislabelled) of errors here and how they relate to the processing rates. E.g.. “…because higher processing rates will risk missing more species whilst lower reduces false-negative error at the expense of increased computational time”

Line 96 – “In more details”, reword to “In more detail” or “Specifically. …”

Line 102-104 – “This metric is therefore very sensitive to detection because few frames have a maximum number of individuals but also to classification if misclassification occurs on the few important frames.” This sentence structure is a bit unclear.

Line 106 – Reword “it appears that by combining fine model tuning with a wisely chosen post processing strategy, there is a potential for optimizing video treatment models to maximize efficiency and accuracy of biodiversity metrics, (i.e. quantifying the true number of organisms).

Methods – Clear methodology described generally well – simulating 10 videos under different conditions and calculating biodiversity indices on the raw video (“manual”) vs. processed videos with different scenarios (“AI”) to compare differences. I do think 10 replicates per condition seems quite low and would like to see some justification for this.

Line 117 – “case study” perhaps?

Line 120-121 – a bit unclear. I think “contrasting abundance and temporal abundance variation” flows better

Line 121 on – very long sentence with some typos. End after “… presence slots.” And reword the next section.

Line 156 – I do think 10 replicates seems quite low. In most simulation studies testing model performance, we typically see N = 100 – 1000 replicate datasets per condition to reduce simulation error. However, I don’t know if this study restricted to N = 10 because videos are very expensive (computationally) to simulate? I’d like to see some justification for why 10 replicates (and not more).

Line 163 – “was done using probability” – more specifics would be good here.

Line 197 – change “we note” to “we assume”, to make it clear its an assumption involved in your simulation, unless it is a fact/result, in which case note is fine.

202 – reword “in an unsure class” to “to an unsure class”. I also think “uncertain” is a better word than unsure unless unsure is used to be consistent with the original paper of Villon et al 2020.

220 – delete eventually

Results – interesting results, clearly presented although a bit wordy and would benefit from shorter sentences.

Line 286 – 288 – long sentence makes results confusing to read, insert comma or break up

Figures

Good figures in general. They clearly show the benefits of post-processing thresholds by showing ground truth species diversity and richness appear to never be recovered regardless of fps and cla_acc in figure 1 (where no processing used I believe).

What is cla_acc? I assume classification accuracy, but add to figure caption.

Discussion – I appreciate the clear guidance in the start of the discussion on processing rates for detection.

Line 365 – this section would benefit from an example of a reasonable processing rate when fast animals are not included

Line 373 – when you mention 53 false positives it would be helpful to have a reminder here of the number of fish and the number of true positives for each condition. (53 / 10000 fish is low but 53 / 100 fish is high)

Line 405 – when mentioning misclassification, it would be beneficial to provide some guidance on how to deal with these situations / correct biases e.g. misclassification models

Line 450 – I now see you do comment on the above here, but it would still benefit from expanding on this in line 405

Reviewer #2: Overall comments:

I think this is a really interesting contribution that stands to save other practitioners substantial time as they attempt to fine tune the implementation of their models in field settings. There are two points that I would like to see clarification on:

1- How the variations in community composition, time thresholds, etc were chosen for the video survey simulations. I would imagine that these factors could vary substantially depending on the community of interest; the authors say that they are “representative” of observed terrestrial and aquatic communities, but it would be helpful to understand what communities are being referenced to get a sense of how broadly applicable these really are.

2- Based on observational results I’m aware of, I would expect that there would be a tendency toward underestimation of species richness, contrary to the results in the simulation. The authors note that even in their best case model, overestimation of richness was driven largely by the false positives, which distributed classification errors evenly among classes due to the assumptions in their simulated models (line 390); in reality, these errors tend to be biased: more common classes have lower error than rare classes, and classes with particular characteristics are more likely to be confused with the background than others—which the authors note (395-405), but assume that this would still lead to a net overestimation. However, data from real world models demonstrate that model accuracy reductions lead to underestimations of species richness via omissions of rare classes (https://arxiv.org/pdf/2408.14348v1). Other assessments of model outputs show a disproportionate impact of confidence thresholding on rare classes, improving precision substantially but at a cost of recall (Willi et al: https://besjournals.onlinelibrary.wiley.com/doi/full/10.1111/2041-210X.13099). This point needs to be addressed in the discussion, if not in the fundamental model assumptions.

Line notes:

Abstract:

31- This sentence could use clarification—on my initial read, I believed that only the videos were simulated, not both the video and the deep learning pipelines

33- Include the relative precision and recall in this sentence—on initial read, it seemed self-evident that models that maximized recall would come at a cost of low precision; now I understand that this sentence actually refers to the model that maximizes both.

35- While raising confidence thresholds will remove false positives, it will also lead to false negatives… how is this threshold set? Only preserve classifications with 0.99 confidence?

Intro:

55- “hence defaulting on the perspective to understand processes of changes” – please clarify, the wording is a bit awkward

71- I think it’s more appropriate to say that these are two of the primary tasks that computer vision models perform, among others (segmentation being a major third task, for instance)

74- The long tail problem is not exclusive to multiclass detectors, this is a generic problem for detection models generally. However, it is true that performance is improved by separating these tasks and removing unneeded/distracting context from classification (e.g., Gadot et al. 2024: https://doi.org/10.1049/cvi2.12318)

As a side note, your citation in this line is not in the same format as the rest of your citations.

85- Deep learning is an umbrella term for a suite of approaches in artificial intelligence that attempt to mimic human learning capacity. Computer vision models are a major branch of deep learning but by no means the only one—generative networks, natural language processing etc are also deep learning.

Methods

116- The phrasing is a bit awkward, I recommend renaming this section

117- citation?

128- Reference for how these were chosen? I can see these varying substantially among different animal communities; would it not make more sense to vary these systematically in a given range for each value so that these are more generalized?

171- How were the frames subsampled for the levels where not all frames were processed—the first frame in a sequence? Randomly chosen?

182- Do you mean that false negatives are ignored?

187- This doesn’t track with my experience of the behavior of detection models, false positives are often clustered/systematic due to the presence of some type of feature in the background that can be confused with an object of interest. I think it is worth noting.

194- This also does not track with the actual behavior of classification models. Rare classes will typically have a higher misclassification rate than common ones, and classes that have less visual differentiation from each other will have higher confusion rates than classes with more visual differentiation from each other. I can see how trying to mimic this behavior would both increase computational complexity due to the specific error magnitudes being somewhat idiosyncratic to each model depending on how it was trained and structured vs the real world distribution of species, but this will have an impact on understanding how your results transfer to applications and is worth addressing why this was done.

Discussion

379- if this is true and the same objects in the background are consistently triggering as false positives, then could stationarity be used as a post processing check (assuming that the camera is static)? IE, checking the distribution of bounding boxes across frames and removing boxes using a threshold of size, /location/number of frames. I realize this goes beyond the scope of your present analysis, but it would be interesting to know if this has been attempted.

429- first pipeline/ second pipeline is a bit unclear here, it reads as if the moderate model with postprocessing had worse errors than the perfect detector with no postprocessing, but I think you intend to say the opposite?

444- This does track with other results from the literature which indicates that setting classification thresholds high does help accuracy generally, but particularly of rare classes (e.g., Willi et al: https://besjournals.onlinelibrary.wiley.com/doi/full/10.1111/2041-210X.13099), but it comes at a cost of reducing recall. For some rare classes, this could result in nondetection—I think this needs to be addressed.

455—classification accuracy also tends to be lower in multiclass detectors, so it is recommended to separate these steps with the present technology (see previous reference to Gadot et al. 2024)

**Do you want your identity to be public for this peer review?** For information about this choice, including consent withdrawal, please see our Privacy Policy

Reviewer #1: No

Reviewer #2: No

---

## [Author Response · Author response to Decision Letter 1]

27 May 2025

We complied at best to all the remarks and answered each of the following questions, rewriting our previous submission, addressing in particular the grammar/language issues and the related work. For an easy reading of the following answers, the authors’ responses are in blue font, lines cited below refer to the tracked changes version.

Comments to the Author

Reviewer #1: General comments – I enjoyed reading this manuscript. A straightforward study processing simulated videos in different ways to calculate biodiversity metrics and compare them, to make some clear recommendations on post-processing for accurate measurements (e.g. 1 frame per second). In general, the study will be strengthened by situating the findings more in the relevant literature considering AI error rates and ecological metrics. I also think N = 10 videos per condition is a small number, although note the patterns between conditions do seem clear even with this sample size.

Thank you for the insightful comments, which have helped us improve our article, especially through comparison with previous studies on AI applied to biodiversity assessment (see detailed answers below)

While 10 simulations may seem limited, the results were indeed consistent across replicates as shown in the figures, where standard deviations around the mean (points) were barely visible. We have added this precision in the text (l. 378).

Introduction - Nice intro motivating the need for underwater video surveys, and the necessity/application of AI processing. There is a reasonable link made between AI errors and downstream effects on biodiversity indices, but, the intro would significantly benefit from a clearer explanation of how errors arise and are handled. A few unclear sentences to be reworded.

Thank you for your suggestions, we proposed a clearer explanation on errors issues and clarified sentences as requested.

Line 50 – 53. Long sentence and a bit unclear. Add comma “without impact, but”. Can you clarify what is meant by structure vs. composition?

We rephrased to clarify the limit of eDNA.

Line 57. Community not communities

Done

Line 82 – Refer to this post-processing as thresholding to be consistent with the existing literature and expand on why this limits errors or cite a source which does so e.g. Improving the integration of artificial intelligence into existing ecological inference workflows - Cowans - Methods in Ecology and Evolution - Wiley Online Library which links AI error rates to ecological inference

Post-processing was changed to thresholding and we added the suggested relevant reference.

Line 89 – “The selected frame processing rate affects the number of images analyzed and hence the likelihood to detect all species for a given model performance.” – perhaps also add considerations of time/processing efficiency here.

We changed this line to take into account the calculation time.

Lines 90 - 93 – This would be improved if you could clarify the types (false negative, false positive) and sources (animal missed, animal mislabelled) of errors here and how they relate to the processing rates. E.g.. “…because higher processing rates will risk missing more species whilst lower reduces false-negative error at the expense of increased computational time”

We clarified this point and now the section read: “Increasing the processing frame rate is expected to reduce false negatives from the detection model (more frames will increase the chance to catch an individual) and hence allow censusing more species and more individuals. Meanwhile, an increasing processing rate is also expected to increase the number of false positives from both detection and classification models resulting in biased species richness and species abundances.”

Line 96 – “In more details”, reword to “In more detail” or “Specifically. …”

Corrected.

Line 102-104 – “This metric is therefore very sensitive to detection because few frames have a maximum number of individuals but also to classification if misclassification occurs on the few important frames.” This sentence structure is a bit unclear.

We rephrased to : “This metric is highly sensitive to detection, as only a few frames typically contain the maximum number of individuals, hence false negatives prevent from censusing all individuals. It is also sensitive to classification, since misclassifications (i.e. false negative) in those key frames decrease the number of individuals from a given species seen simultaneously.”

Line 106 – Reword “it appears that by combining fine model tuning with a wisely chosen post processing strategy, there is a potential for optimizing video treatment models to maximize efficiency and accuracy of biodiversity metrics, (i.e. quantifying the true number of organisms).

We deleted this unclear sentence.

Methods – Clear methodology described generally well – simulating 10 videos under different conditions and calculating biodiversity indices on the raw video (“manual”) vs. processed videos with different scenarios (“AI”) to compare differences. I do think 10 replicates per condition seems quite low and would like to see some justification for this.

Line 117 – “case study” perhaps?

Done

Line 120-121 – a bit unclear. I think “contrasting abundance and temporal abundance variation” flows better

Done

Line 121 on – very long sentence with some typos. End after “… presence slots.” And reword the next section.

Done

Line 156 – I do think 10 replicates seems quite low. In most simulation studies testing model performance, we typically see N = 100 – 1000 replicate datasets per condition to reduce simulation error. However, I don’t know if this study restricted to N = 10 because videos are very expensive (computationally) to simulate? I’d like to see some justification for why 10 replicates (and not more).

We have started analysis with 10 simulations to save computation time (still more than 2 months with a Dell Precision 7550 laptop with an Intel® Core™ i7 processor and 62 GiB RAM). We found that the variability of all studied metrics across these 10 replicates was very low as illustrated on figures where standard deviation is barely visible.

We added “The number of replicates was limited to 10 to keep computation duration low.” and state that inter-replicates variability was low in the Results section.

Line 163 – “was done using probability” – more specifics would be good here.

We've removed this sentence, the explanations of how each model has been implemented are in the text (l. 192 - 199 - 208).

Line 197 – change “we note” to “we assume”, to make it clear its an assumption involved in your simulation, unless it is a fact/result, in which case note is fine.

We reformulated this sentence.

Line 202 – reword “in an unsure class” to “to an unsure class”. I also think “uncertain” is a better word than unsure unless unsure is used to be consistent with the original paper of Villon et al 2020.

Done, we keep the word “unsure” to be consistent with the paper of Villon et al 2020.

Line 220 – delete eventually

Done

Results – interesting results, clearly presented although a bit wordy and would benefit from shorter sentences.

Line 286 – 288 – long sentence makes results confusing to read, insert comma or break up

We changed to: “The Jaccard similarity index between actual species composition and those estimated after automated processing was constant at 0.5 except for 36 models with a processing rate of 0.25 or 0.5 frames per second, for which the Jaccard similarity was between 0.32 and 0.49 (Fig. S2)”.

Figures

Good figures in general. They clearly show the benefits of post-processing thresholds by showing ground truth species diversity and richness appear to never be recovered regardless of fps and cla_acc in figure 1 (where no processing used I believe).

What is cla_acc? I assume classification accuracy, but add to figure caption.

We have added details of the shortcuts used in the figures in the captions for each figure.

Discussion – I appreciate the clear guidance in the start of the discussion on processing rates for detection.

Line 365 – this section would benefit from an example of a reasonable processing rate when fast animals are not included

We moved this sentence further and we added “In such contexts, using a lower processing rate (e.g. 0.5 fps) does not compromise species detection, while significantly reducing the number of analyzed frames and the number of false positives, thereby reducing the proportion of error in biodiversity estimates.” (l. 415).

Line 373 – when you mention 53 false positives it would be helpful to have a reminder here of the number of fish and the number of true positives for each condition. (53 / 10000 fish is low but 53 / 100 fish is high)

We added the number of visible fish for each sub-sampling.

Line 405 – when mentioning misclassification, it would be beneficial to provide some guidance on how to deal with these situations / correct biases e.g. misclassification models

We added “One way to limit errors for these species is to apply a filter based on the size of the detected objects.”

Line 450 – I now see you do comment on the above here, but it would still benefit from expanding on this in line 405

Done

Reviewer #2: Overall comments:

I think this is a really interesting contribution that stands to save other practitioners substantial time as they attempt to fine tune the implementation of their models in field settings. There are two points that I would like to see clarification on:

Thank you for the insightful comments, which have helped us improve our article.

1- How the variations in community composition, time thresholds, etc were chosen for the video survey simulations. I would imagine that these factors could vary substantially depending on the community of interest; the authors say that they are “representative” of observed terrestrial and aquatic communities, but it would be helpful to understand what communities are being referenced to get a sense of how broadly applicable these really are.

The group typology was based on our experience with reef fishes, which vary in abundance, gregariousness, and mobility (e.g., schooling surgeonfish, paired butterflyfish, solitary barracuda). Similar patterns are seen in terrestrial vertebrates monitored with remote cameras (e.g., foxes vs. mouflons). Simulating more groups was limited by computational constraints. We added sentences accordingly in the manuscript.

2- Based on observational results I’m aware of, I would expect that there would be a tendency toward underestimation of species richness, contrary to the results in the simulation. The authors note that even in their best case model, overestimation of richness was driven largely by the false positives, which distributed classification errors evenly among classes due to the assumptions in their simulated models (line 390); in reality, these errors tend to be biased: more common classes have lower error than rare classes, and classes with particular characteristics are more likely to be confused with the background than others—which the authors note (395-405), but assume that this would still lead to a net overestimation. However, data from real world models demonstrate that model accuracy reductions lead to underestimations of species richness via omissions of rare classes (https://arxiv.org/pdf/2408.14348v1). Other assessments of model outputs show a disproportionate impact of confidence thresholding on rare classes, improving precision substantially but at a cost of recall (Willi et al: https://besjournals.onlinelibrary.wiley.com/doi/full/10.1111/2041-210X.13099). This point needs to be addressed in the discussion, if not in the fundamental model assumptions.

While rare classes are often more prone to misclassification, species with distinctive features can still be accurately identified. We chose not to model non-random error structures in our simulations to keep computational demands reasonable, we explained in more detail in the Methods section. We added references that have documented such uneven misclassification patterns. While omissions of rare species at the frame level may lead to underestimations, our analysis focuses on aggregated biodiversity indices across many frames, where this effect can be reduced or even reversed. We added a few sentences in the discussion (l. 396).

Line notes:

Abstract:

31- This sentence could use clarification—on my initial read, I believed that only the videos were simulated, not both the video and the deep learning pipelines

Everything was simulated in this work, because a real pipeline on simulated video would have meant to reconstruct video and work from these. This is not yet feasible. Therefore, the sentence was modified accordingly: “... using simulated remote videos of fish communities and simulated 14,406 automated processing pipelines”.

33- Include the relative precision and recall in this sentence—on initial read, it seemed self-evident that models that maximized recall would come at a cost of low precision; now I understand that this sentence actually refers to the model that maximizes both.

We added the precision, recall and accuracy in the sentence “However, even near-perfect detection (both recall and precision of 0.99) and identification (accuracy of 0.99) models”

35- While raising confidence thresholds will remove false positives, it will also lead to false negatives… how is this threshold set? Only preserve classifications with 0.99 confidence?

To clarify this point, we added this precision: “using a confidence threshold approach (i.e. to discard most erroneous predictions while also discarding a smaller proportion of correct predictions)”

But to fully answer your question, this is explained on line 219 in the method. Basically, the post-processing mimics the approach of Villon (2020). However, we did not simulate a confidence threshold for each species, rather for a given proportion p of FP considered as unsure, there was P/5 TP also considered as unsure.

Intro:

55- “hence defaulting on the perspective to understand processes of changes” – please clarify, the wording is a bit awkward

We changed for “ do not census all species and do not accurately estimate abundance”

71- I think it’s more appropriate to say that these are two of the primary tasks that computer vision models perform, among others (segmentation being a major third task, for instance)

Done

74- The long tail problem is not exclusive to multiclass detectors, this is a generic problem for detection models generally. However, it is true that performance is improved by separating these tasks and removing unneeded/distracting context from classification (e.g., Gadot et al. 2024: https://doi.org/10.1049/cvi2.12318)

As a side note, your citation in this line is not in the same format as the rest of your citations.

Thank you for the relevant reference, we added it and we updated the citation format.

85- Deep learning is an umbrella term for a suite of approaches in artificial intelligence that attempt to mimic human learning capacity. Computer vision models are a major branch of deep learning but by no means the only one—generative networks, natural language processing etc are also deep learning.

We agree and corrected to “computer vision models”

Methods

116- The phrasing is a bit awkward, I recommend renaming this section

We changed for “Simulating video-based community surveys”

117- citation?

We added several citations, Catalan et al, 2023 proposed a model with 20 classes, Burgi et al. 2024 proposed a model with 19 classes plus one ‘other’ class with various species, for terrestrial mammals Rigoudy et al., 2023 proposes a classification model for 26 common mammals in Europe. So the number of classes chosen corresponds to what is done in the literature.

128- Reference for how these were chosen? I can see these varying substantially among different animal communities; would it not make more sense to vary these systematically in a given range for each value so that these are more generalized?

The group typology was partly based on our experience with temperate and tropical reef fishes which in addition to marked differences in total abundance exhibit diverse combinations

---

## [Decision Letter · Decision Letter 1]

6 Jun 2025

Dear Dr. Fleuré,

Thank you for submitting your manuscript to PLOS ONE. After careful consideration, we feel that it has merit but does not fully meet PLOS ONE’s publication criteria as it currently stands. Therefore, we invite you to submit a revised version of the manuscript that addresses the points raised during the review process.

We look forward to receiving your revised manuscript.

Kind regards,

Tzen-Yuh Chiang

Academic Editor

PLOS ONE

Journal Requirements:

Reviewers' comments:

Reviewer's Responses to Questions

**Comments to the Author**

Reviewer #1: All comments have been addressed

2. Is the manuscript technically sound, and do the data support the conclusions?

Reviewer #1: Yes

3. Has the statistical analysis been performed appropriately and rigorously?

Reviewer #1: Yes

4. Have the authors made all data underlying the findings in their manuscript fully available?

Reviewer #1: Yes

5. Is the manuscript presented in an intelligible fashion and written in standard English?

Reviewer #1: Yes

Reviewer #1: Line 23 – video based surveys are

Line 27/80 – “However, the impacts on models' performances and usage on assessment of biodiversity metrics on videos remains unknown.” Or better say “is yet to be assessed” to reflect wording in the into.

Line 29 – The sentence is long and unclear – either say “14,406 simulated automated processing pipelines” or reword to “Using simulated remote videos of fish communities processed with 14,406 simulated automated pipelines, …”

Line 43 – consistently tracking (swap order)

Line 55 – change developing to “development of”

Line 79 – “false positives (regions of image without any individuals but detected as containing one).”

Line 97 – “is yet to be assessed”

Line 119 – Intro ends a bit abruptly. Add one line at end of intro to summarise how. E.g. “We achieve this using simulated remote videos of fish communities and simulated 14,406 automated 32 processing pipelines, and calculating error rates and biodiversity metrics.”

Line 116 –“The number of replicates was limited to 10 due to high computational cost.” Better say “to balance computational cost with reproducibility”, to show your results were reproducible even with 10 repeats.

**Do you want your identity to be public for this peer review?** For information about this choice, including consent withdrawal, please see our Privacy Policy

Reviewer #1: No

---

## [Author Response · Author response to Decision Letter 2]

9 Jun 2025

Reviewer #1:

Thank you for your careful reading and helpful corrections of grammatical errors, which improved the clarity of the manuscript.

Line 23 – video based surveys are

Done

Line 27/80 – “However, the impacts on models' performances and usage on assessment of biodiversity metrics on videos remains unknown.” Or better say “is yet to be assessed” to reflect wording in the into.

Done

Line 29 – The sentence is long and unclear – either say “14,406 simulated automated processing pipelines” or reword to “Using simulated remote videos of fish communities processed with 14,406 simulated automated pipelines, …”

Done

Line 43 – consistently tracking (swap order)

Done

Line 55 – change developing to “development of”

Done

Line 79 – “false positives (regions of image without any individuals but detected as containing one).”

Done

Line 97 – “is yet to be assessed”

Done

Line 119 – Intro ends a bit abruptly. Add one line at end of intro to summarise how. E.g. “We achieve this using simulated remote videos of fish communities and simulated 14,406 automated 32 processing pipelines, and calculating error rates and biodiversity metrics.”

Done

Line 116 –“The number of replicates was limited to 10 due to high computational cost.” Better say “to balance computational cost with reproducibility”, to show your results were reproducible even with 10 repeats.

Done

---

## [Decision Letter · Decision Letter 2]

18 Jun 2025

Impact of deep learning and post-processing algorithms performances on biodiversity metrics assessed on videos

PONE-D-25-14344R2

Dear Dr. Fleuré,

We’re pleased to inform you that your manuscript has been judged scientifically suitable for publication and will be formally accepted for publication once it meets all outstanding technical requirements.

Kind regards,

Tzen-Yuh Chiang

Academic Editor

PLOS ONE

Additional Editor Comments (optional):

Reviewers' comments:

Reviewer's Responses to Questions

**Comments to the Author**

Reviewer #1: All comments have been addressed

2. Is the manuscript technically sound, and do the data support the conclusions?

Reviewer #1: Yes

3. Has the statistical analysis been performed appropriately and rigorously?

Reviewer #1: Yes

4. Have the authors made all data underlying the findings in their manuscript fully available?

Reviewer #1: Yes

5. Is the manuscript presented in an intelligible fashion and written in standard English?

Reviewer #1: Yes

Reviewer #1: (No Response)

**Do you want your identity to be public for this peer review?** For information about this choice, including consent withdrawal, please see our Privacy Policy

Reviewer #1: No

---

## [Editor Report · Acceptance letter]

PONE-D-25-14344R2

PLOS ONE

Dear Dr. Fleuré,

I'm pleased to inform you that your manuscript has been deemed suitable for publication in PLOS ONE. Congratulations! Your manuscript is now being handed over to our production team.

Kind regards,

on behalf of

Dr. Tzen-Yuh Chiang

Academic Editor

PLOS ONE